# The Reliability and Medical Students’ Appreciation of Certainty-Based Marking

**DOI:** 10.3390/ijerph19031706

**Published:** 2022-02-02

**Authors:** Špela Smrkolj, Enja Bančov, Vladimir Smrkolj

**Affiliations:** 1Faculty of Medicine, University of Ljubljana, 1000 Ljubljana, Slovenia; enja.bancov12@gmail.com (E.B.); vladimir.smrkolj@gmail.com (V.S.); 2Department of Obstetrics and Gynaecology, University Medical Centre, 1000 Ljubljana, Slovenia

**Keywords:** certainty-based marking, confidence-based learning, undergraduate medical education, online exam, self-evaluation

## Abstract

Certainty-Based Marking (CBM) involves asking students not only the answer to an objective question, but also how certain they are that their answer is correct. In a mixed method design employing an embedded approach with a quasi-experimental design, we have examined the use of CBM during a 5-week Gynaecology and Obstetrics course. The study was conducted as a non-mandatory revision exam with two additional questionnaires on Moodle. Majority of students perceive CBM as fair (78%) and useful (94%). Most students would immediately want CBM to be used for revision exams, but more practice would be needed for CBM to be used in graded exams. The lowest self-evaluation of knowledge was mostly seen by worst (less than 70% Accuracy) and best achievers (more than 90% Accuracy); the worst achievers probably have knowledge gaps, and the best achievers probably correctly guessed at least one question. Our findings conclude that CBM does not discriminate any learner type (*p* = 0.932) and does not change the general distribution of the exam scores, since there is no significant differences between Certainty-Based Score (*M* = 80.4%, *SD* = 10.4%) and Accuracy (*M* = 79.8%, *SD* = 11.1%); *t*(176) = 0.8327, *p* = 0.4061. These findings are widely applicable, as learner type study models are used extensively in education. In the future, larger samples should be studied and the implementation of CBM on question types other than MCQ should be investigated.

## 1. Introduction

The differentiation between knowledge and luck on exams has been puzzling researchers for many decades. Some of the older studies, such as Coombs et al. [1], Hevner [2] and Soderquist [3] already tried to incorporate certainty or confidence into exams. Gardner-Medwind has extensively incorporated CBM into revision exams at University College London, where he had over 9000 sessions of CBM use. He optimised the certainty bonus factors and the scoring of certainty [4]. CBM upgrades Multiple Choice Questions (MCQ), so that students must indicate how certain they are in their answer. The more reliable is their self-evaluation of their knowledge, and the greater reward they receive (CBM Bonus) [5,6]. When using CBM, one must indicate the degree of certainty in the given answer; the scale is not defined exactly, but many authors, including Gardner-Medwind, use a 3-point scale [6]. The descriptions of the scale are *low-mid-high*, which correspond to 50%-67%-80% certainty. The scoring then follows according to a scoring table, in which high certainty receives a maximum award or penalty, and low certainty has no penalty if wrong, but receives the least points if the answer is correct. There are then multiple indicators which describe a student’s performance on the exam; the percentage of correct answers (Accuracy), overall CBM Bonus (CBM Bonus) and the Certainty-Based Score. The CBM Bonus (given in %) shows a student’s deviation from the standard curve in terms of the self-evaluation of certainty, while the Certainty-Based Score is the sum of Accuracy and CBM Bonus. Automatic analysis of the answers at each certainty level can also give students feedback, whether they have been over or under confident at each certainty level [6,7]. Considering all of the above, the use of CBM should enable both the examiner and the student to more precisely evaluate the student’s knowledge [8].

However, the integration of CBM in the curriculum is only present in a small number of faculties, which widens the discrepancy between students’ and teachers’ appreciation of the use of CBM and its actual availability in the curriculum [9]. To yield results, CBM must not be integrated solely as a type of examination, but rather as a part of Confidence-based learning, which is a philosophical idea—that knowledge is not only the correctness of answers but also the person’s certainty. That is manifested as the difference between “I know” and “I think” [10]; a difference most important in medicine, as an error can result in tragedy. In fact, medical workers are encouraged to consult in case of doubt, which can only be possible if a person can discriminate between true knowledge and uncertain knowledge. The above-described philosophy should be taught to students throughout the entire study program, as it indirectly changes the student’s study behaviour towards a performance improvement. This indirect effect is thought to result from the student’s self-regulated learning—the student will isolate the topics where she or he lacks knowledge and focus on them [11,12]. Our experience with medical students is that they are aware of the fact that their knowledge is incomplete but they find it hard to quantify and isolate the topics where they have the least knowledge. Moreover, it is common that they have conceptual misunderstandings; consequently, they are not aware of the fact that they lack knowledge [13,14,15]. To support this, Hendriks et al. [9] concluded that CBM is greatly appreciated by students; however, more accessible options to integrate CBM should be developed, which has now been done [16]. Regular revision exams can be used to enhance medical education, as they promote better retention of information. This phenomenon is known as the testing effect [17]. Furthermore, the phenomenon can be further expanded by providing more detailed feedback, although this is yet to be confirmed by solid evidence [18]. Again CBM is promising a self-assessment tool to regularly give students the ability to test their knowledge and receive more detailed feedback. However, the feedback received from CBM is not as detailed as the one given by a teacher [15,19,20,21,22]. In contrast, computer based learning methods show no overt effect when compared to traditional study methods [23]. Moreover, the factor of luck in MCQ is a known design problem and when not enough care is taken to actively discourage the random guessing, exams become unreliable [24,25,26,27,28,29]. CBM was designed to tackle the issues presented above—identifying students’ conceptual misunderstandings, identifying students’ areas for improvement and actively discouraging random guessing [30,31,32].

It should be noted that the students’ performance on all exams may differ due to their adaption to the type of the exam. This could be connected to the student’s learner style, and effort should be made to not discriminate against either learner style if learner style models are concluded to be valid. There are many learning style models and schemas described in the literature; however, they have been followed with controversy and questionable validity. One systematic review tried to evaluate the 14 most influential models, whereas they listed over 70 models that have been developed [33]. One of the most commonly used models is the VARK model, developed by N. Fleming and C. Mills in 1992 [34]. It differentiates between four basic learner types (modalities): Visual, Auditory, Read/Write and Kinesthetic. Despite the VARK model’s debated validity it is used widely between students, teachers and researchers [35,36,37,38]. Between different question types, MCQ was shown to be least sensitive to learner style [39]. For CBM to be useful in the curriculum it must maintain this trait.

In this article we are trying to tackle the implementation stage of the CBM addition to the curriculum. We are evaluating the reliability of CBM in terms of student learning types and their self-reported study efforts; moreover, we aim to gather their feedback on implementing the CBM in the curriculum, to deduce if students would like to have CBM used in exams. Considering the above, we have set the following research questions: Is CBM reliable in terms of students’ learner types? and Do students appreciate the use of CBM in curriculum?

## 2. Method

### 2.1. Study Design

The mixed method design employed was an embedded approach with a quasi-experimental design. The quantitative method was quasi-experimental between-subjects approach utilizing non-manipulated independent variables based on a closed-ended pretest questionnaire. Qualitative data were collected as a post-test questionnaire [40,41,42]. Independent variables were learner type, number of times the student rehearsed the topic, when the student started learning and the student’s self-evaluated knowledge level, whereas dependent variables were Accuracy, CBM bonus and Certainty-Based Score. To sum up, the study was conducted as a non-mandatory revision exam with two additional questionnaires: one before (quantitative data) and one after the exam (quantitative and qualitative data).

### 2.2. Data Collection

Both the CBM exam and the questionnaires were built on the Moodle open-source learning platform [16]. We have chosen Moodle, as it is a free learning platform, which is accessible to any faculty, moreover it readily supports CBM exams as a free plug-in. Before the study was conducted, the students were given a short description of the study, were told what CBM is and how to use it. Firstly, a questionnaire about the learner style and study efforts was given to the students. The learner style part of the first questionnaire was a translated VARK questionnaire [43], while the study efforts part asked the students to determine when they started learning, how many times they rehearsed the subject matter and how they perceive their knowledge of the subject. As a base for the learner style questionnaire we have used an English VARK questionnaire version 8.01 [44]. The VARK questionnaire consists of 16 MCQ, where each answer is characteristic for one of the four modalities of VARK theory. The student’s modality is determined by the highest number of answers that correspond to that modality. We have had the questionnaire translated by a C1 certified English speaker from English into Slovene. The translated questionnaire was independently back-translated. Minor changes to the initial translation were made after the back translation. The validity of the questionnaire was approved by expert committee, which consisted of three members. The first member was an expert on the topic of interest and the methodology; the second was the forward translator and the third was the backward translator. The translated questionnaire was then pilot tested with a small group of students (*n* = 32) as recommended by Tsang et al. [45]. After students in our study finished the first questionnaire, the CBM exam was unlocked and students were given 20 min to solve 20 MCQ questions with CBM and only one correct answer. After the CBM exam they were not immediately shown their achievement, but they answered the second questionnaire. The second questionnaire asked them about the difficulties they had when using CBM, whether their achievement was impaired because of CBM and if they see the use of CBM as fair. In addition they had an option to leave general feedback in the form of a text-box field. All together the duration of the study was 60 min. The translated questionnaire is available in the Appendix A.

### 2.3. Sample

We have examined the use of CBM during a 5-week Gynaecology and Obstetrics course at the Faculty of Medicine at the University of Ljubljana. The students who participated in the study were fifth-year medical students (*n* = 89) from four different rotations (groups). The mean age of students was 23.04 years (*SD* = 0.49 year). Three quarters of students were females (74%), the rest were males (26%). They all consented to participate and provide data for the study.

### 2.4. Data Analysis

Student performance was compared based on their questionnaire response using one-way ANOVA, where more than three categories per variable were present, and using the *t*-test, where only two categories per variable were present. Only the students who completed the questionnaires before and after the CBM exam were included in the analysis of the effect of learner type and students’ self reported study efforts on the CBM performance. In the analysis of the distribution of scores between Accuracy and Certainty-Based Score, the students who did not complete the questionnaires were also included. Quantitative data from the second survey (feedback) were analysed with basic descriptive statistics. Qualitative data were organized and examined. To aid in the analysis of the qualitative data we have developed some tags (arguments that have occurred in the feedback) and tried to extract the most important ones. Statistical analysis was performed in OriginPro 2021 software.

## 3. Results

### 3.1. Participation

The participation rate in the study was 62% (89 out of 143 students). Whoever did not want to participate in the study was given the opportunity to solve the revision exam without CBM or with CBM, while their data were not included in the study. Out of 89 students who participated in the study, 45% (40 students) did not solve the questionnaires before and after the CBM revision exam.

### 3.2. Student Previous Experience

Students’ responses to the questionnaire before the CBM exam are presented in Figure 1. Surprisingly, 29% of students had no previous experience with online exams, moreover only one student was previously familiar with CBM.

Most students were of the visual learner type, followed by auditory and kinesthetic learner types. Moreover, the ratio of learner types was similar in all the groups that we included in the study. A quarter of students said they learned regularly, while about a half started learning two weeks before the study. The date of the study had been known in advance as well as the dates of all the other study activities and the topics that were on the CBM revision exam.

### 3.3. Students’ Feedback

The majority of students indicated that they perceive CBM as fair (78%) and useful (94%) as seen in Figure 2.

Twenty-five students also described their experience in the text box field in the questionnaire. From the feedback we received, the most popular concern is that someone who showed less absolute knowledge (Accuracy) can receive a better grade, some however thought that CBM is useful, but it cannot be directly implemented, as it requires a mental shift. Three students were concerned that the CBM Bonus is too big, and it should be reduced, their argument is that currently overly self-critical students would be negatively discriminated. One student pointed out that the questions must be stated clearly to avoid ambiguity, as that could result in students being less certain in their answers. Several students reported that including CBM in non-revision exams would increase their stress-level at these exams. Students were generally satisfied with CBM experience and have stated it would be very beneficial to be used at least as a revision tool. Some (15 students) reported they had been able to identify knowledge gaps using the automatic feedback that was given at the end of CBM (after the second questionnaire). All of the data from the open type questions (text box) are available upon enquiry.

### 3.4. Reliability of CBM in Terms of Learner Types and Students’ Self-Reported Study Efforts

There is no statistically significant link between students’ learner types and their achievement in any of the CBM parameters (Accuracy, CBM Bonus, Certainty-Based Score), as seen in Table 1.

However, when comparing students’ answer accuracy with their average CBM mark, the deviation from the standard curve is clearly seen (Figure 3). In general, the students’ achievements are distributed equally above and below the standard curve; however, top-achievers in terms of accuracy (above 90%) have mostly scored below the standard curve. Similarly, the bottom-achievers in terms of accuracy (below 70%) have scored generally below the standard curve. The best achievers in terms of CBM mark are those in the middle range of accuracy (70–90%).

Comparing the grade distribution of Accuracy and Certainty-Based Score (Accuracy + CBM Bonus) in (Figure 4) yields that there is no significant differences between Certainty-Based Score (*M* = 80.4%, *SD* = 10.4%) and Accuracy (*M* = 79.8%, *SD* = 11.1%); *t*(176) = 0.8327, *p* = 0.4061.

However, the Certainty-Based Score has more negative skewness (offset to the right) than Accuracy, which suggests that the grades shifted to the right; however, both the minimum and the maximum grade are lower in terms of Certainty-Based Score than in terms of Accuracy. Although the Certainty-Based Score distribution seems to be shifted to the right, only 60% of students received positive CBM Bonus. Consequently, it is obvious that the distribution has been roughly preserved; however, the students themselves have changed their relative order in many cases.

## 4. Discussion

Increased use of computer-based multiple choice questioning in the COVID pandemic has enabled us to upgrade content using CBM. The pioneering work of Gardner-Medwin and colleagues [1,2,3,4,5,6,30] has been exploited to support diverse educational goals. As CBM has never been incorporated in the curriculum at our Faculty, we have considered it is high time we researched the reliability and medical students’ appreciation of CBM.

### 4.1. Students’ Previous Experience

Most of the students who participated in the study were already familiar with Moodle as an online study and exam platform, which suggests that there was some use of online learning tools already before the study. However still 29% of students were new to online exams. This highlights that there is a substantial inequality between students in terms of experience with technology, which must be reduced to minimum, before these online tools are used in curriculum [46]. It must be noted that our study was conducted during the first year of COVID-19 pandemic, therefore the first group in the study was new to online studying, while later groups were more accustomed to it. Similarly, this was discussed by Aguilera-Hermida [47] where she found an increased use of online educational technology during the pandemic by almost 45%. Students’ increased experience with online study tools is beneficial for the inclusion of CBM into the curriculum. During the pandemic, online learning tools have had a crucial role in filling the gaps and ensuring uninterrupted education, as was discussed by Chatterjee and Chakraborty [48]. Therefore, it is our duty to change the study programs in a way to be more future proof, which certainly includes online learning platforms, some of which already include CBM, as has been demonstrated in our study with an example from Moodle.

### 4.2. Students’ Feedback

The feedback we received from students has shown that the use of CBM could potentially be successful and well received if enough preparation would have been done. Most importantly, the implementation must be done gradually for students who are not familiar with the concept, as was mentioned by few students in their feedback. It is a reasonable request from students, as only one student in our study had previous experience with CBM. Therefore it is not a surprise that many students reported in their feedback that they only think CBM should be used for revision exams. Almost all students (94%) saw CBM as a valuable tool for revision, which is also one of the key aspects that the original author of CBM, Gardner-Medwin, has promoted [6,30]. In contrast, Schoendorfer and Emmett [49] reported that only around 60% of students appreciated the benefits of CBM, moreover 67% of students in their study have reported that it took them more time to solve exam and consequently 30% of students saw CBM as a “waste of time thinking about certainty”. Similarly, Hendriks et al. [9] reported that 32.7% of students were against the use of CBM. Perhaps the difference between our students’ high appreciation of CBM compared to the previously discussed studies is due to our study being conducted during pandemic, when students were deprived of many other forms of feedback. Considering what students in our study have written in the feedback, we can observe that many of them felt positive about the auto-generated feedback about their over or under-confidence. While students were encouraged to reflect on their knowledge throughout the process of solving the CBM exam, the benefits of the process would have been lesser if they had not received the feedback per certainty level and topic. Per their responses, it is only with that information that allows them to improve their knowledge where it gaps. Furthermore, Hendriks et al. [9] reported similar conclusions—52.2% of students in their study considered auto-generated feedback useful. These observations answered our research question, which asked if students appreciated CBM.

We have anticipated that most of the students rehearsed the study matter at least once, but it was unexpected that 18% of them rehearsed it three times. Because the duration of the course was five weeks, and the study was performed during the third week of the course, it is highly doubtful that they have rehearsed all the topics three times. More so since it had been made clear that this study would not affect their course grade in any way and that rather than studying hard for the CBM exam, they should not change their study plan and take the CBM exam as a revision exam. This is definitely a weakness of our study, which could be tackled with a larger sample. Most (60%) of our students evaluated their knowledge to be good, 4% evaluated it to be bad and 36% evaluated it to be excellent. From the feedback we received, we concluded that we should devise the self-evaluation of knowledge per topic, as it is easier to evaluate it in that way, rather than questioning about the general knowledge, which is hard to quickly and accurately self-evaluate.

### 4.3. Reliability of CBM in Terms of Learner Types and Students’ Self-Reported Study Efforts

We have not established a statistically significant difference between the achievements of students with different learner types. In the light of our first research question we can conclude that CBM is reliable in terms of students’ learner types. Although there is no consensus about whether learner types are indeed valid, its use is unquestionably widespread, which makes our finding important [35,36,37,38]. We can also confirm that the findings from Holley and Jenkins [39], which suggest that the MCQs have high reliability in terms of learner types. Furthermore we found that this trait is preserved when using MCQ in conjunction with CBM.

As we anticipated, there is no statistical difference between the achievements of students with or without the use of CBM. This is due to Gardner-Medwin [4]’s analysis of several thousand CBM revision exams and his optimization of the standard curve and the bonus factor, to yield good psychometric reliability and decrease the inflation of grades (reduces the predictive value of the exam). As mentioned in the Introduction, the CBM should not be implemented as an isolated tool, but rather as a part of Confidence-based learning philosophy taught through the entire program. In this case students would be more familiar with the concept, its benefits and the platform, as suggested by the following authors Gardner-Medwin [6], Agrawal et al. [11] and Larsen et al. [17]. Otherwise, the exam results would not be reliable as some students who are less agile with the concept of CBM would be negatively discriminated against. Further study should be conducted to determine how much practice is needed by the students to be able to solve exams without being hindered by the complexity of the exam platform or the CBM.

Surprisingly, we have not established a statistical difference in achievements of students between the number of times they rehearsed or when they started learning. We can only speculate about the cause of that, perhaps we should ask students to evaluate their knowledge at each topic, then their self-evaluations could be more objective. In further studies, more participants should be included; consequently, the analysis of their achievements in a group will be more reliable, yielding results that could be—if CBM could differentiate students with more profound knowledge—better than just the accuracy of answers. However, it is already visible from the results in Figure 3 and Figure 4 that while the distribution of scores between Accuracy and Certainty-based Score is relatively unchanged, the students’ in-group order must have been changed. That means that the use of CBM most likely will not alter the general score distribution of a group, as that is a function of the difficulty of questions and general knowledge in a group, but will rather reorder the students in-group to correct for their good or bad confidence, similarly noted by Hendriks et al. [9]. This is important because CBM cannot be used to replace good and valid questions in exams; CBM can only enhance their discrimination power. We have observed that the worst self-evaluators of knowledge were the worst and best-achievers. The best achievers probably at least partially scored highly due to luck; however, the bottom achievers were probably unfamiliar with the topics and perhaps had certain knowledge misconceptions. Wu et al. [22] and Gardner-Medwin [4] observed the same phenomenon that the best self-evaluators of knowledge were mid-achieving students.

## 5. Conclusions

This study explored the use, appreciation and reliability of CBM with medical students. CBM is not a new tool, but its benefits have not been fully utilised yet. Moreover, it is easily setup with widely used learning platforms such as Moodle. Our findings showed that most students perceive CBM as a positive addition to their study, but some are still sceptical about whether CBM can be more than just a revision tool. Adding certainty ratings to MCQ answers seems to reinforce formative testing with better feedback. It also offers deeper insight into the successful delivery of online course content, identifying areas for improvement of teaching and content delivery as well as exam question design. Secondly, our findings conclude that CBM does not discriminate against any learner type (based on VARK questionnaire) and it does not change the general distribution of the exam scores. In fact, our findings are relevant to the whole educational community, since the use of learner type models is widespread. In future studies, a larger sample should be studied and it would also be wise to examine the implementation of CBM on question types other than MCQ.

### 5.1. Usefulness of CBM

CBM in a formative assessment improves student course appreciation, presumably by helping students to evaluate their mastery level and identify misconceptions. Certainty-based learning inclusion in formative assessments may impinge on exam designs and course evaluations, hence representing an important and useful assessment tool for teaching staff as well [5,9,10,11]. During the COVID pandemic, the study of medicine has been largely distance-based, with the exception of clinical exercises. For this reason, it is especially important that students have a tool such as CBM available in the learning process, so that they can gain more reliable knowledge, which they can then put into daily practice in a clinical setting. Moreover we must encourage students to aid their studies with internet based methods, consequently future lock-downs will be less stressful for them and much easier to overcome [50]. Perhaps most important is the enhancement of learning with certainty-based marking self-tests [6]. It is argued that such self-tests should be available to students to explore in private, making mistakes and finding strengths and weaknesses without submitting any data to the view of teachers.

### 5.2. Limitations

Using a larger sample size, we could reduce random errors, due to dishonesty in answers to the questionnaires. Moreover our study did not take into account the possibility of academic dishonesty such as cheating on the revision exam. This is a design flaw that is inherent due to the nature of the observations.To explain better: we evaluated online exams which students solved from home, and if it involved a supervision of some kind it would make the study biased. Some students also did not want to participate in the study. If there is some underlying reason as to why they did not want to participate, such as their low self-confidence, that would make our sample unrepresentative and our findings less valid. Moreover, students expressed their perception in the middle of pandemics. In such conditions, students could have increased fear or stress, which could negatively affect some of the students’ performances.

## Figures and Tables

**Figure 1 ijerph-19-01706-f001:**
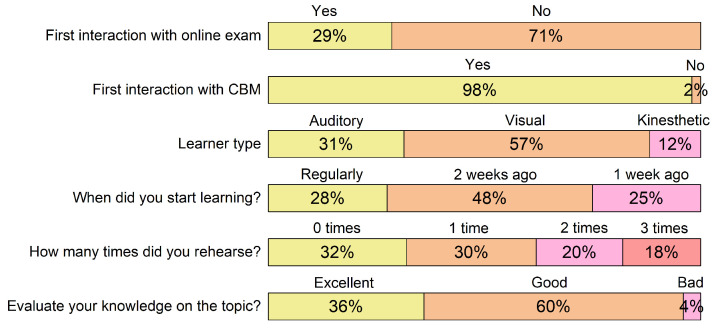
Students’ responses to the questionnaire before the CBM exam (n=49). The labels on the left represent the question or a series of questions that were given in the questionnaire. The text above the stacked bars represents the possible answers, while the width of the bar represents the portion of students who selected that answer. The data for *Learner type* questions has been analysed and only the determined proportions for each learner type are represented.

**Figure 2 ijerph-19-01706-f002:**
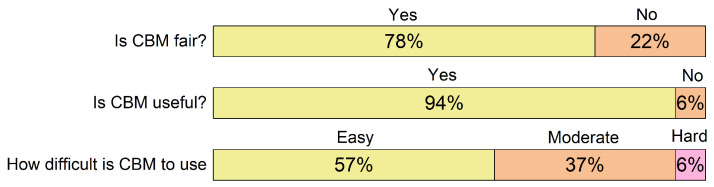
Students’ responses to the questionnaire after the CBM exam (n=49). The labels on the left represent the question or a series of questions that were given in the questionnaire. The text above the stacked bars represents the possible answers, while the width of the bar represents the portion of students who selected that answer.

**Figure 3 ijerph-19-01706-f003:**
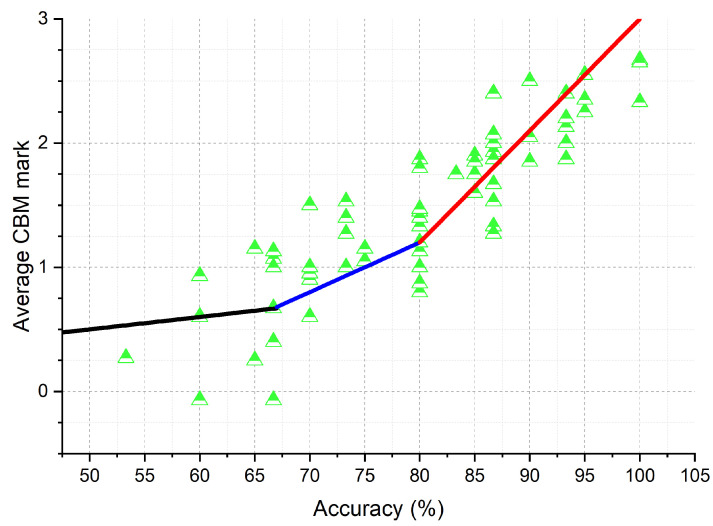
Average CBM mark (=All points/number of questions) versus Accuracy shows the deviation from the standard curve, which represents random guessing.

**Figure 4 ijerph-19-01706-f004:**
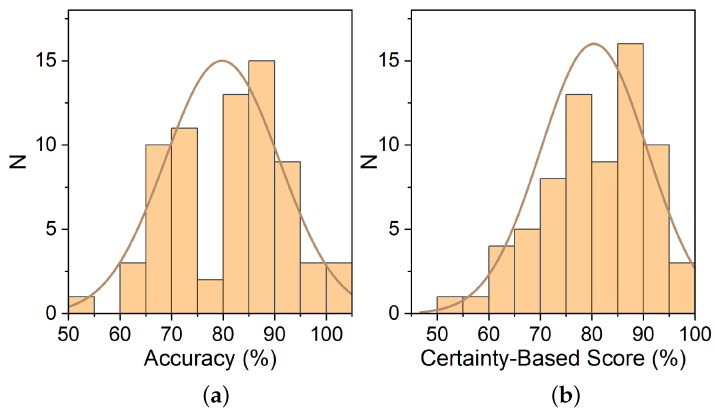
(**a**) Distribution of Accuracy scores. (**b**) Distribution of Certainty-Based Score.

**Table 1 ijerph-19-01706-t001:** Results of mean-comparison tests between different student’s qualities (Learner type, Learning start, Rehearsal times, Self-evaluated knowledge) and performance indicators (Accuracy, CBM Bonus and Certainty-Based Score).

Independent Variable	*p*-Value		
	**Accuracy**	**CBM Bonus**	**Certainty-Based Score**
Learner type	0.900	0.711	0.932
Learning start	0.512	0.940	0.453
Rehearsal times	0.593	0.862	0.423
Self-evaluated knowledge	0.901	0.492	0.718

## Data Availability

The data that supports the findings of this study are available from the corresponding author, Š.S., upon reasonable request.

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
