# Peer review of "The Reliability and Medical Students’ Appreciation of Certainty-Based Marking"

_ijerph, 2022, doi:10.3390/ijerph19031706_

Round 1

Reviewer 1 Report

This is an interesting and significant study. It contributes to the understanding how the CBM can be of great benefit to medical students in comparison to the traditional MCQ. In the rationale and research design, the author/s make relevant and clear ideas to the research topic. I'd recommend that the authors can elaborate about the number of participants and whether the facts that they did or did not compete the questionnaires before and after the CBM would affect the validity of the study. 

Author Response

Response to reviewers’ comments

We thank the reviewer for his/her valuable insights and constructive comments that have contributed to a better quality of our manuscript. Below are our responses in italics:

Reviewer 1

This is an interesting and significant study. It contributes to the understanding how the CBM can be of great benefit to medical students in comparison to the traditional MCQ. In the rationale and research design, the author/s make relevant and clear ideas to the research topic. I'd recommend that the authors can elaborate about the number of participants and whether the facts that they did or did not compete the questionnaires before and after the CBM would affect the validity of the study.

We thank the reviewer for pointing out that we haven’t stated which students were included in the analysis. Where the data from the questionnaires was required for the analysis only the students who completed both questionnaires were included. Whereas in the comparison of the grade distribution there is no data required from the questionnaires, therefore also the rest of students’ data were used. Considering this, we believe that the validity of the study was not affected. This information is now added to the manuscript in the section »2.4 Data analysis«.

Reviewer 2 Report

The paper lacks a clear explanation of the central concept of CBM. The introduction should clearly define it and provide a brief review of the literature. While a rationale for the research is provided, it is unconvincing in regards to further application of CBM in this context.

The method and analysis are clearly presented and sufficiently rigorous. As is reported about the students, I am also sceptical about the usefulness of CBM beyond revision purposes. A fuller explanation and perhaps some qualitative data on why students felt it was useful would help with this.

Author Response

Response to reviewers’ comments

We thank the reviewer for his/her valuable insights and constructive comments that have contributed to a better quality of our manuscript. Below are our responses in italics:

Reviewer 2

The paper lacks a clear explanation of the central concept of CBM. The introduction should clearly define it and provide a brief review of the literature. While a rationale for the research is provided, it is unconvincing in regards to further application of CBM in this context.

We thank the reviewer for directing us to provide a clearer explanation of CBM in the Introduction. We have added more details about CBM, moreover we included more references to better support the central concept of CBM. In the introduction we have reduced the number of subsections and we tried to make a more natural connection between the parts of Introduction (where the problem, rationale and objective are stated and where the rest of the issues are addressed in greater depth). These modifications are present in the section »1. Introduction«.

The method and analysis are clearly presented and sufficiently rigorous. As is reported about the students, I am also sceptical about the usefulness of CBM beyond revision purposes. A fuller explanation and perhaps some qualitative data on why students felt it was useful would help with this.

We have added more qualitative data that we have collected during our study. This data has mostly come from the text box field in the last questionnaire. We tried to include as many relevant comments as possible. The added qualitative data is present in section “3.3 Students’ feedback”.  Moreover, we have added a separate subsection in “Discussion”: “4.4. Usefulness of CBM”. We have also given extra care to highlight the benefits or limitations of CBM throughout the discussion.

Reviewer 3 Report

From a global point of view, the manuscript "The reliability and medical students' appreciation of Certainty-Based Marking" presents potentially interesting research on the use of CBM in medical students.

From my point of view from an educational research perspective, several issues should be reviewed in depth in order to proceed to a new revision:
- The introductory section appears divided into too many subsections, in some cases without natural connection. The manuscript should differentiate between an introductory section (where the problem, rationale and objective are stated) and a theoretical framework section (where the rest of the issues are addressed in greater depth than they currently are).
- In the Materials and Methods section (I suggest renaming it Method), a subsection explaining the research methodology followed (type of study, nature, design, etc.) should be included at the beginning. Then, in the section describing the sample, the context in which the study was carried out should be described, as well as the mean and deviation of age, the number of men and women and any other characteristic of the participants that could provide relevant information. Next, it is necessary to rename subsection 2.2 to "Data collection" since the design should be incorporated in the first subsection I named above. In addition, the process of translation and validation of the instruments used should be presented in this "Data Collection" subsection.
- The discussion section should incorporate more references to other studies, making comparisons, especially in subsections 4.1 and 4.2.
- Finally, the conclusions and contributions of the study should be explained in greater depth, in addition to pointing out its limitations.

Author Response

Response to reviewers’ comments

We thank the reviewer for his/her valuable insights and constructive comments that have contributed to a better quality of our manuscript. Below are our responses in italics:

Reviewer 3

From my point of view from an educational research perspective, several issues should be reviewed in depth in order to proceed to a new revision:

- The introductory section appears divided into too many subsections, in some cases without natural connection. The manuscript should differentiate between an introductory section (where the problem, rationale and objective are stated) and a theoretical framework section (where the rest of the issues are addressed in greater depth than they currently are).

We thank the reviewer for directing us to provide a clearer explanation of CBM in the Introduction. We have added more details about CBM, moreover we included more references to better support the central concept of CBM. In the introduction we have reduced the number of subsections and we tried to make a more natural connection between the parts of Introduction (where the problem, rationale and objective are stated and where the rest of the issues are addressed in greater depth). These modifications are present in the section »1. Introduction«.

- In the Materials and Methods section (I suggest renaming it Method), a subsection explaining the research methodology followed (type of study, nature, design, etc.) should be included at the beginning. Then, in the section describing the sample, the context in which the study was carried out should be described, as well as the mean and deviation of age, the number of men and women and any other characteristic of the participants that could provide relevant information. Next, it is necessary to rename subsection 2.2 to "Data collection" since the design should be incorporated in the first subsection I named above. In addition, the process of translation and validation of the instruments used should be presented in this "Data Collection" subsection.

We have followed the reviewer’s suggestion and renamed section “Materials and Methods” into “Method”. We have changed the structure of the section “Method”, so that it is in this order: Study design, Data collection instruments, Sample and Data Analysis. We have better described our Study design and have indicated our in-/dependant variables. The description of data collection instruments was improved with a detailed description of the translation and validation of the instruments used. We have also given arguments regarding the choice of the learning platform. In the description of the Sample we included the demographic description of the sample (Age, Gender). The other relevant information about the sample has already been presented – the students are 5th year Medical students. These modifications are present in the section »2. Method«.

- The discussion section should incorporate more references to other studies, making comparisons, especially in subsections 4.1 and 4.2.

- Finally, the conclusions and contributions of the study should be explained in greater depth, in addition to pointing out its limitations.

Considering the reviewers comments we have greatly improved our discussion as a whole. We have added more references and have strived to compare our findings with more relevant and recent studies. We have also tried to isolate the reasons why some of our findings were different from other studies. To better describe our contributions, we have made explicit connection between our discussion and our research questions. Moreover, we have added a subsection in “Discussion” evaluating the usefulness of CBM and another subsection discussing the limitations of our study (subsections “4.4 Usefulness of CBM” and “4.5 Limitations”). Finally, we have improved our conclusion by directly stating our conclusions and our contributions to the whole educational community.

Reviewer 4 Report

Dear authors in my opinion, the study needs some changes that I expose next.

INTRODUCTION.-

- The abstract should be improved for publication of the paper in this journal. Include a sentence about the design and underscore the scientific value added of your paper.

- The authors divide the "introduction" section into very short sections. The introduction and theoretical background needs to be adjusted so as to provide a deeper background on the subject matter. Add more recent/relevant and diverse references.

- In section 1.5 (the aim of study) the authors do not define the objective of the research nor do they formulate research questions (diffuse).

METHODOLOGY.-

The Material and Methods section should be improved. Section 2 can be broken down into: research objective or questions, data collection instruments, sample, data analysis, etc.

- Authors should clearly state the objective or questions of their research.

- Why these research instruments and tools? On what theory are the applied questionnaires based?

- Authors must show the applied questionnaire in an appendix and explain its relevance (theory underlying the questions, validation process, etc.).

RESULTS.-

- Figure 1 is not well explained and difficult to understand.

- In the results section the authors make comments that they later repeat in the discussion (they are not results).

DISCUSSION.-

- The discussion is vague (especially section 4.1). In the discussion, the authors must interpret the results obtained in the light of the research questions formulated and articulate it to the theoretical framework related to the research (in this case CBM).

LIMITATIONS.-

- It is advisable that the authors include in their work a section on the main limitations of their study.

FORMAL ISSUES.-

- Authors should strive to be more synthetic in their work. Sometimes authors repeat arguments already given.

- Figure 1 needs to be improved (not seen)

Author Response

Response to reviewers’ comments

We thank the reviewer for his/her valuable insights and constructive comments that have contributed to a better quality of our manuscript. Below are our responses in italics:

Reviewer 4

INTRODUCTION.-

- The abstract should be improved for publication of the paper in this journal. Include a sentence about the design and underscore the scientific value added of your paper.

We thank the reviewer for directing us to provide a better abstract, in which we included a sentence about the design of our study, moreover we have explicitly stated the contributions of our paper to the whole educational community.

- The authors divide the "introduction" section into very short sections. The introduction and theoretical background needs to be adjusted so as to provide a deeper background on the subject matter. Add more recent/relevant and diverse references.

We have strived to improve our Introduction and include a clearer explanation of CBM. We have added more details about CBM, moreover we included more references to better support the central concept of CBM. In the introduction we have reduced the number of subsections and we tried to make a more natural connection between the parts of Introduction (where the problem, rationale and objective are stated and where the rest of the issues are addressed in greater depth). These modifications are present in the section »1. Introduction«.

- In section 1.5 (the aim of study) the authors do not define the objective of the research nor do they formulate research questions (diffuse).

Following the suggestion of the reviewer we have added two clear research questions in the last paragraph of the section “Introduction”. Moreover, we tried to make the aim/objective of the study more defined.

METHODOLOGY.-

The Material and Methods section should be improved. Section 2 can be broken down into: research objective or questions, data collection instruments, sample, data analysis, etc.

 Authors should clearly state the objective or questions of their research.

Why these research instruments and tools? On what theory are the applied questionnaires based?

Authors must show the applied questionnaire in an appendix and explain its relevance (theory underlying the questions, validation process, etc.).

Because we have included the aim of the study and research questions in the last paragraph of the “Introduction”, we have decided it would not be necessary to include it again in the section “Method”. We have changed the structure of the section “Method”, so that it is in this order: Study design, Data collection instruments, Sample and Data Analysis. We have better described our Study design and have indicated our in-/dependant variables. The description of data collection instruments was improved with a detailed description of the translation and validation of the instruments used. We have also given arguments regarding the choice of the learning platform. In the description of the Sample we included the demographic description of the sample (Age, Gender). The other relevant information about the sample has already been presented – the students are 5th year Medical students. These modifications are present in the section »2. Method«.

The theory of the learner types and the questionnaires is given in the section “Introduction” (VARK model). After consideration we have followed the reviewer’s suggestion and have briefly described the VARK questionnaire in the subsection “2.2 Data collection instruments”. The original English VARK questionnaire is available online (also added in the list of references), whereas our translated questionnaire is available upon enquiry and not in the appendix, as it is in Slovene language.

RESULTS.-

- Figure 1 is not well explained and difficult to understand.

Considering the reviewer’s comment, we have split Figure 1 into two figures: one with results from the questionnaire before the CBM exam (new Fig 1), and the other with results from the questionnaire after the CBM exam (new Fig 2). We have also tried to improve the description below the Figures, which now describes the elements of Figure 1 and Figure 2 better.  

- In the results section the authors make comments that they later repeat in the discussion (they are not results).

We have carefully read through the “Results” section and have moved certain comments to section “Discussion”.

DISCUSSION.-

- The discussion is vague (especially section 4.1). In the discussion, the authors must interpret the results obtained in the light of the research questions formulated and articulate it to the theoretical framework related to the research (in this case CBM).

LIMITATIONS

- It is advisable that the authors include in their work a section on the main limitations of their study.

Considering the reviewers comments we have greatly improved our discussion as a whole. We have added more references and have strived to compare our findings with more relevant and recent studies. We have also tried to isolate the reasons why some of our findings were different from other studies. To better describe our contributions, we have made explicit connection between our discussion and our research questions. Moreover, we have added a subsection in “Discussion” evaluating the usefulness of CBM and another subsection discussing the limitations of our study (subsections “4.4 Usefulness of CBM” and “4.5 Limitations”). Finally, we have improved our conclusion by directly stating our conclusions and our contributions to the whole educational community.

FORMAL ISSUES.-

- Authors should strive to be more synthetic in their work. Sometimes authors repeat arguments already given.

- Figure 1 needs to be improved (not seen)

These issues have been addressed in previous answers.

Round 2

Reviewer 2 Report

Thank you to the authors for taking onboard my earlier comments. The introduction has been improved with a robust definition and explanation of CBM.

It is useful that the qualitative open-text comments from students have been added. These allow the reader to see the reactions to CBM and while I am still uncertain of the use of CBM beyond revision, the potential advantages are more clearly stated.

Author Response

We thank the reviewer for his/her valuable insights and constructive comments that have contributed to a better quality of our manuscript.

Reviewer 3 Report

The version presented by the authors has been substantially improved. However, I still find some important issues that need to be resolved for the article to be publishable.

I fail to properly understand the information in subsection 2.1. Are the authors referring to a mixed-method study? If so, what type, with what sequencing? The design should be clearly explained and the corresponding bibliographic references should be included where the structure of the selected design can be contrasted.

Regarding section 2.2: in the first place, it should be renamed "Data collection", since not only the instruments are presented but also the collection process. Secondly, it should be made explicit from which language the questionnaire has been translated into. Thirdly, the authors comment that "The validity of the questionnaire was approved by expert committee", but in what way? The authors should describe the process followed and what type of experts were involved. Fourth, there is one issue that particularly concerns me. The authors state that "The translated questionnaire is available upon enquiry". However, the questionnaire needs to be shown as an attachment to this manuscript, not upon request. Otherwise, the study lacks rigor.

Finally, subsections 4.4 and 4.5 should not be presented in the discussion but in the conclusions section. In fact, they would substantially reinforce this section of conclusions.

Author Response

The version presented by the authors has been substantially improved. However, I still find some important issues that need to be resolved for the article to be publishable.

I fail to properly understand the information in subsection 2.1. Are the authors referring to a mixed-method study? If so, what type, with what sequencing? The design should be clearly explained and the corresponding bibliographic references should be included where the structure of the selected design can be contrasted.

Answer: "We thank the reviewer for pointing out our vague study design description. We have put more effort in explicitly describing our study design and the type of sequencing, furthermore we have included three additional references that provide further details about study designs, their strengths and limitations and an example of a study utilising similar design. We have also updated abstract to reflect the improved section “2.1 Study design”."

Regarding section 2.2: in the first place, it should be renamed "Data collection", since not only the instruments are presented but also the collection process. Secondly, it should be made explicit from which language the questionnaire has been translated into. Thirdly, the authors comment that "The validity of the questionnaire was approved by expert committee", but in what way? The authors should describe the process followed and what type of experts were involved. Fourth, there is one issue that particularly concerns me. The authors state that "The translated questionnaire is available upon enquiry". However, the questionnaire needs to be shown as an attachment to this manuscript, not upon request. Otherwise, the study lacks rigor.

Answer: "Considering the reviewer’s comments, we have renamed section 2.2. into “2.2. Data collection”. Moreover, we have explicitly stated that the questionnaire has been translated from English to Slovene. More importantly we have included a short description of the expert committee and its members. Most importantly we have added the translated questionnaire (in Slovene, as the English version is already cited in the section References (reference 44)) into Supplementary materials. We hope that these changes will make our study more rigorous."

Finally, subsections 4.4 and 4.5 should not be presented in the discussion but in the conclusions section. In fact, they would substantially reinforce this section of conclusions.

Answer: "We have followed the reviewer’s suggestion and moved subsections 4.4 and 4.5 to the section “5. Conclusions”. We believe that the conclusion is now more strengthened with the inclusion of the before-mentioned subsections."

Reviewer 4 Report

I congratulate the authors for their work. It is evident that a great deal of effort has been put into improving the article.

Author Response

(The authors gave the same response as above.)
